# Antibacterial, Antibiofilm, and Antioxidant Activity of 15 Different Plant-Based Natural Compounds in Comparison with Ciprofloxacin and Gentamicin

**DOI:** 10.3390/antibiotics11081099

**Published:** 2022-08-12

**Authors:** Ali Pormohammad, Dave Hansen, Raymond J. Turner

**Affiliations:** 1Department of Biological Sciences, Faculty of Science, University of Calgary, Calgary, AB T2N 1N4, Canada; 2C-Crest Laboratories Inc., Montreal, QC H1P 3H8, Canada

**Keywords:** antibacterial agents, antibiotics, biofilms, plant-based compounds, mechanism, natural compounds, herbal, essential oils, cannabinoids, *Caenorhabditis elegans*, toxicity, antioxidant

## Abstract

Plant-based natural compounds (PBCs) are comparatively explored in this study to identify the most effective and safe antibacterial agent/s against six World Health Organization concern pathogens. Based on a contained systematic review, 11 of the most potent PBCs as antibacterial agents are included in this study. The antibacterial and antibiofilm efficacy of the included PBCs are compared with each other as well as common antibiotics (ciprofloxacin and gentamicin). The whole plants of two different strains of *Cannabis sativa* are extracted to compare the results with sourced ultrapure components. Out of 15 PBCs, tetrahydrocannabinol, cannabidiol, cinnamaldehyde, and carvacrol show promising antibacterial and antibiofilm efficacy. The most common antibacterial mechanisms are explored, and all of our selected PBCs utilize the same pathway for their antibacterial effects. They mostly target the bacterial cell membrane in the initial step rather than the other mechanisms. Reactive oxygen species production and targeting [Fe-S] centres in the respiratory enzymes are not found to be significant, which could be part of the explanation as to why they are not toxic to eukaryotic cells. Toxicity and antioxidant tests show that they are not only nontoxic but also have antioxidant properties in *Caenorhabditis elegans* as an animal model.

## 1. Introduction

Antimicrobial resistance is a serious public health concern, especially for people with underlying diseases such as cancer, diabetes, AIDS, and other chronic diseases, as well as immunocompromised patients (reviewed by Tanwar et al.) [1]. Additionally, during the COVID-19 pandemic, mixed infections and multidrug resistance bacteria (MDR) rates have increased, which can cause life-threatening problems [2]. Hence, antibiotic resistance has reached a critical stage that has led us into an antimicrobial resistance (AMR) era. Consequently, the world health organization (WHO) has published a list of bacteria for which new antibiotics are urgently needed [3].

Natural products/essential oils/plant-based natural compounds (PBCs) are suggested to have strong potential in approaches to formulate novel antimicrobials [4,5,6]. PBCs not only have the advantage of being relatively inexpensive to produce but also often have limited toxicity and side effects to the host. Moreover, most PBCs have other benefits to the host such as anti-inflammatory and anticancer features, in addition to boosting the immune system [6,7,8,9].

Previous studies have primarily focused on one PBC or their isoforms/precursors to survey antibacterial potency against a specific strain of bacteria [10,11,12,13]. However, there are more than 50 different PBCs reported to have natural antibacterial activity (reviews [14,15]). Yet, there is still an important question about which PBCs can be effective against a specific infection-causative microorganism and what concentrations are required. A direct comparison of all previous results carried out in different conditions with different strains to answer the question is impractical. Furthermore, when a study is designed to indicate a specific PBC’s antibacterial efficacy, publication bias for that PBC is possible.

To address these concerns, relevant publications on the antimicrobial potential of various PBCs were systematically searched to identify the most potentially effective antibacterial agents. Based on our systematic review, 11 defined PBCs as the antibacterial agent were chosen to be included in our comparative study in addition to 4 crude extracts from cannabis plant material. The antibacterial and antibiofilm efficacy of the included PBCs were compared with each other; moreover, we compared their antimicrobial and antibiofilm potency with two common antibiotics (ciprofloxacin and gentamicin) against six WHO priority-listed bacteria for the Gram-negative bacteria *Pseudomonas aeruginosa*, *Escherichia coli*, *Klebsiella pneumoniae*, *Proteus mirabilis*, and *Acinetobacter baumannii,* and the Gram-positive bacteria *Staphylococcus aureus*. Additionally, since high pure cannabidiol (CBD) and tetrahydrocannabinol (THC) had promising antibacterial and antibiofilm features against all six bacteria, we also explored whole plant crude extracts from two different strains of *Cannabis sativa*. The most common antibacterial mechanisms were explored to understand the antibacterial mechanism of selected PBCs. Finally, toxicity and antioxidant tests for assessing the safety of the compounds were carried out in a *Caenorhabditis elegans* (*C. elegans*) model.

This is the first systematic comparison of literature for in vitro and in vivo antimicrobial and toxicity experiments carried out under identical conditions to multiple PBCs and multiple bacterial strains to reduce bias and obtain a valid decision towards the use of different PBCs’ antibacterial efficacy and safety.

## 2. Results

### 2.1. A Systematic Review on Plant-Based Natural Compounds (PBCs) as Antibacterial Agents

There are numerous natural components with antibacterial properties [6]. Moreover, different analogues of the same PBCs have different antibacterial features. Thus, for selecting the most effective natural components to explore in our study, a systematic review of previous publications was carried out. As expected, results from different studies were highly varied and the minimal inhibitory concentration (MIC) ranges were quite broad for the same PBCs. For example, extremely variable MICs (<1 to >2000 µg/mL) were reported for CBD. CBD was seen to be effective against Gram-positive bacteria in the majority of the studies [16,17,18,19,20], yet CBD was not effective against Gram-negative bacteria in some studies [16,17,18]. However, this may depend on the preparations, as other CBD extractions were effective against Gram-negative bacteria as well [19,20]. Likewise, a wide range of MIC values were reported for the antibacterial potency of other PBCs: resveratrol 10–>1000 µg/mL; thyme essential oils 20–128,000 µg/mL; carvacrol <0.4–1700 µg/mL; tea tree oil 1–16,000 µg/mL; cinnamaldehyde 2–1000 µg/mL; nerolidol 0.1–0.4; coumaric acid 1.67–>2000 µg/mL. The variabilities are, in part, due to the inconsistency of bacterial species and strains tested, their culture methods and the source and processing of the PBCs (Appendix A). This makes it difficult to effectively compare efficacies of PBCs and thus this led to our subsequent experiments.

### 2.2. Antibacterial Potency of PBCs in Planktonic Form

For the convenience of using our data for comparisons in the literature, we report all antibacterial and antibiofilm potency of PBCs in two different units (mM and μg/mL). Antibacterial susceptibilities in the main text values are reported in μg/mL units and in the Appendix A reported in mM to facilitate comparisons to other reports. Here, we evaluated the MIC (bacteriostatic), the minimal biocidal concentration (MBC—for bactericidal) and the minimum biofilm inhibition concentration (MBIC). Figure 1 shows the MIC, MBC, and MBIC of 15 different PBCs for the Gram-negative bacteria *Pseudomonas aeruginosa*, *Escherichia coli*, *Klebsiella pneumoniae*, *Proteus mirabilis*, and *Acinetobacter baumannii*, and the Gram-positive bacteria *Staphylococcus aureus*. The PBCs that had better or equal bacteriostatic, bactericidal, or antibiofilm potency in comparison with antibiotics are highlighted in Appendix A. Resveratrol and curcumin, under our experimental conditions, did not show antibacterial efficacy in the selected concentration (i.e., >40,000 μg/mL).

Two antibiotics were chosen as benchmarks to compare with the PBCs under identical experimental conditions. Ciprofloxacin and gentamicin were chosen as they are broad-spectrum antibiotics which are effective against both Gram-positive and Gram-negative bacteria. Moreover, these are two of the most commonly used antibiotics from recent decade(s), and their mechanism of antibacterial activity is well known (see reviews [21,22,23,24]).

### 2.3. Bacteriostatic Potency of PBCs

The MIC values reflect the bacteriostatic activity of the PBCs. Bacteriostatic efficacy of some PBCs were better than ciprofloxacin and gentamicin. Cannabidiol had better bacteriostatic activity (with MIC of (0.026–0.8 μg/mL)) compared to ciprofloxacin MIC (0.5–1 μg/mL) and gentamicin MIC (1–3 μg/mL) against *S. aureus*. Additionally, cinnamaldehyde at MIC of (1.5–6 μg/mL), carvacrol (0.2–3 μg/mL), and thymol (12.5–62 μg/mL) had better bacteriostatic activity against *A. baumannii* in comparison with both antibiotics, ciprofloxacin (31–132 μg/mL) and gentamicin (62–764 μg/mL) (Figure 1 and Appendix A).

### 2.4. Bactericidal Potency of PBCs

Figure 1 shows the MBC values that reflect the bactericidal activity of the PBCs. Like the bacteriostatic activity of the PBCs, bactericidal features of some PBCs were better than ciprofloxacin and gentamicin. Cinnamaldehyde had an MBC of 126.5–131.25 μg/mL. Carvacrol (6.25–25 μg/mL) and Thymol (19.5–65 μg/mL) had better bactericidal activity against *A. baumannii* in comparison with both ciprofloxacin (126–250 μg/mL) and gentamicin (62.5–859 μg/mL). The bactericidal activity of the PBCs compared with antibiotics recorded in mM units are reported in the Appendix A.

### 2.5. Antibiofilm Potency of PBCs

Figure 1 shows the MBIC of 15 PBCs against *P. aeruginosa, S. aureus, E. coli*, *K. pneumoniae*, *P. mirabilis*, and *A. baumannii* grown as a biofilm produced by shear force on the polystyrene surface [25,26,27]. Although MBIC is expected to be similar to the MIC, as seen with most antibiotics, for many non-antibiotic antimicrobials and particularly those that are bacterial static versus bactericidal, the MBIC can be quite different from the MIC. Here, we see that cinnamaldehyde (with a MBIC of 4–25 μg/mL), carvacrol (0.45–6.25 μg/mL), and thymol (9–25 μg/mL) had better biofilm inhibitory activity against *A. baumannii* in comparison to both ciprofloxacin (33–125 μg/mL) and gentamicin (62.5–764 μg/mL). Appendix A shows the biofilm inhibitory activity of the PBCs compared with antibiotics in mM units.

### 2.6. Targeted Treatment of P. aeruginosa, S. aureus, E. coli, K. pneumoniae, P. mirabilis, and A. baumannii with PBCs

Table 1 reports the best PBCs and concentrations towards the six bacteria (*P. aeruginosa, S. aureus, E. coli*, *K. pneumoniae*, *P. mirabilis*, and *A. baumannii*). Overall, CBD and THC had promising antibacterial and antibiofilm features against all six bacteria, especially *S. aureus*. For targeted treatment, cinnamaldehyde, carvacrol, and thymol had acceptable antibiofilm and antibacterial activity against *A. baumannii*. Out of 15 PBCs, thymol was the most effective antibiofilm and antibacterial against *K. pneumoniae*. Cannabidiol and carvacrol had promising antibiofilm and antibacterial against *E. coli.* Against the *P. aeruginosa*, cannabidiol and THC displayed better antibiofilm and antibacterial efficacy. Thymol had the best antibiofilm and antibacterial activity against *P. mirabilis*.

### 2.7. Potential Antibacterial Mechanism of Selected PBCs

#### 2.7.1. Oxidative Stress and Fe-S Complex

Previous studies have demonstrated that antibacterial agents target different pathways to kill bacteria, as discussed in several review papers [28,29], including the disruption of ATP production, DNA replication, breaking down [Fe-S] clusters, the generation of reactive oxygen species (ROS), direct damage to cell membranes, etc. In our study, to understand the antibacterial mechanism of selected PBCs, the most common antibacterial mechanisms were explored together under identical conditions allowing direct comparisons. Oxidative stress and ROS are some of the most common mechanisms for most antibacterial agents [29]. ROS can damage cell membranes, DNA, and cellular proteins, and may lead to cell death [29,30]. Therefore, we measured the hydrogen peroxide (H_2_O_2_) concentrations to sense ROS potential after 1hr exposure of the bacteria with the selected PBCs and ciprofloxacin and gentamycin as an internal control. Cinnamaldehyde and gentamycin produced the same amount of H_2_O_2_ (0.91 ± 0.047 µM and 0.91 ± 0.065 µM, respectively). The same amount of H_2_O_2_ was produced by ciprofloxacin and THC (0.81 ± 0.05, 0.81 ± 0.004 µM, respectively). However, CBD (0.72 ± 0.14 µM) and carvacrol (0.52 ± 0.001 µM) reduced H_2_O_2_ concentration (Figure 2), while the untreated control (bacteria + WR) had quite a close concentration (0.74 ± 0.05). These data show that H_2_O_2_ concentrations in our selected PBCs were almost the same or less than ciprofloxacin, gentamycin, and the untreated group. Therefore, oxidative stress is not the main antibacterial mechanism of these antibiotics [23,24].

The antimicrobial challenge can lead to ROS production [31,32,33,34]. This can occur from: (1) membrane damage, which results in the uncoupling of the electron transport chain and thus increases electron flow to oxygen; (2) the decomposition of respiratory enzymes leading to breaking down [Fe-S] clusters and the subsequent release of Fe^2+^ which can catalyze Fenton reactions. Direct colorimetric assay (Ferene-S) was used to measure Fe^2+^ concentrations after exposure with PBCs. The Fe^2+^ concentration of 1 h exposed bacteria with candidate PBCs and antibiotics were compared with the negative control (treated with PBS) and positive control (10 min boiling to break the cluster). The difference in Fe^+2^ concentration between the treated and negative control was not significant (*p* < 0.23). This shows that PBCs, ciprofloxacin, and gentamycin do not kill via this mechanism.

#### 2.7.2. Membrane Permeability

The specific structure of the bacterial cell wall helps to protect them from the host immune system and different environmental stresses [34]. However, the unique structure of the bacterial cell membrane is targeted by some antibiotics such as polycations and chelators [34]. This feature helps antibiotics to kill the bacteria without affecting the host cells [34]. To survey membrane disruption potency in our selected components, the membrane leakage probe propidium iodide (PI) that binds DNA was used in this study to detect membrane destabilization after 1 h exposure with our PBCs (Figure 3). All our selected PBCs significantly disturbed the cell membrane in comparison with the untreated group (*p* < 0.001). Data show that PBCs directedly targeted the bacterial cell membrane rather than using ROS and the oxidative stress pathway. Previous studies have shown that this is not the main antibacterial mechanism for ciprofloxacin, and gentamycin [23,24]; our results show the same result.

### 2.8. Toxicity of PBCs towards Caenorhabditis elegans

Figure 4 shows the *C. elegans* toxicity of 15 different PBCs in comparison with ciprofloxacin and gentamicin. After treating nematodes with different PBCs, four important viability variables were explored for five days and compared with the untreated group. Some of the PBCs such as THC, cinnamaldehyde, tea tree oil, carvacrol, and coumaric acid had even more growth, motility, regeneration, and population numbers compared with the untreated control group, suggesting that they stimulate the health of the animal. Regeneration started after day four, so this variable is significant within our assessment. For instance, THC had a better regeneration ratio (115–125%) than the control, while gentamicin had a lower regeneration ratio (80–85%) in comparison with the control group.

### 2.9. The Antioxidant Potency of PBCs in Caenorhabditis elegans Model

Many natural products have antioxidant properties by preventing the production of the reactive superoxide radical (O_2_^−^) and general reactive oxygen species (ROS) [35]. Free radical production leads to cell wall lysis, biomolecular damage, the disruption of cells, the activation of the immune system, inflammation, and then finally organ and tissue damage [36,37].

For treated and untreated *C. elegans* groups, the ROS level was measured with the DCFH-DA probe. A noticeable difference between ciprofloxacin and PBCs groups was detected. More specifically, higher fluorescence intensity was observed in the untreated control 18.5 (±1.2) relative fluorescent units (RFU) and ciprofloxacin 20 (±1.4) RFU in comparison with THC 12 (±2.6) RFU, CBD 14 (±1.1) RFU, cannabinoids 12 (±2.9) RFU, and tea tree oil 12.5 (±1.5) RFU (Figure 5). Likewise, Figure 6 shows the O_2_^−^ level by using the DHE probe. The fluorescence intensity of THC 10 (±3.8) RFU, CBD 6 (3.5–9.5) RFU, cannabinoids 7(±4.2) RFU, and tea tree oil 5 (±1.5) RFU were remarkably lower than the control 16.7 (±1.8) RFU and ciprofloxacin 18.6 (±1.2) RFU groups.

Reduced glutathione (GSH) is an important antioxidant and it plays a critical role in protecting against oxidative stress in cells, inflammation and injury [38]. Here, the GSH level of *C. elegans* was measured after exposure to PBCs and antibiotics by the NDA probe. As shown in Figure 7, the fluorescence intensity of THC 662.6 (±2.8) RFU, CBD 746 (±3) RFU, cannabinoids 753(±3.5) RFU, and tea tree oil 729 (±2.4) RFU were considerably higher than the control 312.6 (±0.6) RFU and the ciprofloxacin 172 (±1.6) RFU groups. This suggests the stimulation of GSH antioxidant biosynthesis in the presence of many PBCs, which correlates with the reduced ROS levels (s). Figure 8 shows the general view of PBCs mechanism in bacterial and eukaryotic cells.

## 3. Discussion

PBCs have been used as an alternative treatment from ancient times for a variety of ailments, including infection control [39,40,41,42]. Nowadays, PBCs have regained attention for their availability, diverse structures and limited side effects [42]; particularly, PBCs are considered potential sources of novel antibacterial agent templates [42], especially given the increase in antibiotic resistance and the need for novel antibiotics [43,44]. To address key knowledge gaps in this area, here we investigated PBCs to directly compare efficacy towards various WHO priority pathogens and clarify further their safety as antibacterial agents. We started by carrying out a systematic review of previous publications to identify potential highly effective PBC-based antibacterial agents.

We considered and found that, even for a single species of bacteria, different specific antibacterial susceptibility patterns have been reported for various PBCs (Appendix A). A wide range of MIC/MBC for a specific PBC might be due to different qualities of PCB formulations, dissimilar bacterial strains, and the various experimental conditions used in different studies. Concerns for PBCs include different extraction methods, different PBC isotypes, and different producer companies with different extraction protocols and purities. All these issues lead to the lack of clarity about using PBCs as an antibacterial agent and which are best for what organism.

There are many reports relevant to PBCs’ antibacterial efficacy [45,46,47], yet none give a clear comparative analysis. Thus, based on our initial literature review, we selected 15 PBCs to compare their antibacterial potency against two antibiotics. This generated a large array of data; thus, for the convenience of the reader, and an easier comparison of data, we focus on the most used antibacterial susceptibility method (MIC) in this study [48,49]. Serial dilution MIC determination is recommended by the Clinical and Laboratory Standards Institute (CLSI) for susceptibility testing [48] and the results achieved by some susceptibility methods such as disc diffusion should be confirmed by the MIC method (considered as a reliable reference method) [49,50]. Most of the publications we found used either the MIC or disc diffusion method [20,51]. However, disk diffusion results are affected by many factors, such as pH, composition and the electrolytes of agar media, agar–antibacterial component interactions, humidity, incubation time and temperature [52,53]. Moreover, the diffusion potency of the antibacterial agent has a great impact on the inhibition zone. For example, vancomycin and polymyxins’ poor diffusion affects their inhibition zones, while they have great antibacterial efficacy [52,53]. Thus, disc diffusion reports were excluded from our analysis and only the MIC method was included from this study.

Many natural products have antibacterial features such as extracted biomaterial from fungi [54,55,56] or bacteria [57,58], marine bacteria [59,60,61,62,63,64] and other marine creatures [55,65,66], macrophages [63,67,68], natural peptides [69,70,71], and herbal/plant extracts [72,73,74]. Herbal/plant-based components are of interest because of their availability, diverse structure, and limited side effects [75,76]. A wide range of PBCs could be considered for microbial studies, though we only included the components reported by studies with a promising MIC. Components that were reported by only one or two studies, or MIC > 5000 μg/mL, or with highly variable MICs, were excluded from our study [74]. For instance, *Persicaria pensylvanica* was reported as a promising antibacterial against *S. aureus* (MIC of 7.8–62.5 μg/mL) but was not included because it was reported in only one study. Additionally, access to this plant is limited as it only grows in specific ecological regions [77].

Based on our systematic review, PBCs with the highest potential were selected and included to survey their antibacterial and antibiofilm properties against six pathogen indicator strains under identical conditions. The results identified that highly pure CBD and THC had promising antibacterial efficacy against both Gram-positive and Gram-negative bacteria. In our study, CBD was effective against both Gram-positives and Gram-negatives, similar to a few previous reports [19,20]. Cannabis contains more than 500 ‘bioactive’ components, but THC and CBD are the main constituents of consideration, and most likely the best antibacterial agent candidates [51,78]. At this time, pure cannabinoids are quite expensive, so we performed a simple extraction of whole plant material from two different *Cannabis sativa* locally cultivated strains with two different methods to compare their antibacterial potency with the ultrapure THC and CBD. Our results support previous reports [78,79] where large differences were detected between ultrapure THC/CBD and whole plant crude extracts in their effective antibacterial and antibiofilm concentrations (Figure 1), even though such simple extractions retain psychoactive effects. For instance, the whole plant extractions were not bactericidal in our concentration range (>40,000 μg/mL), while CBD was bactericidal against *E. coli* at a very low concentration (12.5–25 μg/mL). This suggests that the antibacterial feature of cannabis is mostly related to THC and CBD rather than non-cannabinoid constituents [78,79], and that there must be other plant compounds present in the extract that are acting antagonistically.

Antibiotics can be bacteriostatic or bactericidal. The bactericidal potency of an antibiotic is a critical advantage because it can decrease resistance chance as well as bacterial recovery after exposure to the antibiotic [80]. Furthermore, some antibiotics not only have bactericidal activity but also have antibiofilm features to prevent or eradicate persistent infections as well [81]. Persistent infections and the recovery potency of some bacteria after exposing them to some antibiotics are some of the biggest clinical and industrial challenges we face [82]. This is due to antimicrobials being bacteriostatic vs. bactericidal [83]. In this study, most of our selected PBCs had antibacterial and antibiofilm potency in a very close concentration. For instance, carvacrol had a MIC of (125–250 μg/mL) against *K. pneumoniae*; likewise, the MBC and MBIC range were very close (250–300 μg/mL). Therefore, the combined bactericidal and antibiofilm potency of the selected PBCs provides them with excellent potential use in a variety of applications.

Most of the time, the targeted treatment of an antimicrobial can be quite critical, especially when we want to keep the normal healthy flora community [84]. The most effective antibacterial and antibiofilm PBCs and effective concentrations to apply against specific WHO priority strains are recommended in this study (Table 1). Different strain physiologies, cell wall structures, gene expression profiles, the presence of various drug efflux pumps, and other resistance strategies suggest why some agents are effective for a specific bacterial strain, but not for others [85,86].

Previous studies demonstrate that antibacterial agents target different pathways to kill the bacteria, as discussed in several review papers [28,29], including the role of the disruption of ATP production, DNA replication, breaking down [Fe-S] clusters, the generation of reactive oxygen species (ROS), direct damage to cell membranes, etc. However, the antibacterial mechanism of most PBCs remains unclear. A deeper understanding of the molecular mechanism and specific targets could help to increase antibiotic efficacy, decrease host toxicity, and allow in turn the extension to industrial and clinical applications [87]. Our results showed that almost all our selected PBCs follow the same pathway for their antibacterial effects. They mostly target bacterial cell membrane in the initial step, which has been reported by other studies as well [88,89]. The bacterial cell membrane has a far different composition than the eukaryotic cell membranes [90] and since the PBCs were less toxic to eukaryotic cells, this composition difference is a probable for their selectivity and thus safe antibacterial agents. Moreover, our selected PBCs did not significantly induce H_2_O_2_ production, especially carvacrol (reduced levels to less than the untreated control). ROS production is toxic for both bacterial and eukaryotic cells. This might be another reason for the lack of toxicity of PBCs. Consequently, *C. elegance* model was used to survey the toxicity and safety of PBCs.

Important criteria for developing new antimicrobials are the side effects and host cell cytotoxicity of the compound. Many chemicals and materials have strong antibacterial activity, but they also have host cell cytotoxicity [91,92]. For assessing the toxicity of PBCs, a *C. elegans* model was utilized and showed that some of the PBCs would actually increase the worm’s fitness compared with the untreated control group. THC, CBD, cinnamaldehyde, tea tree oil, carvacrol, and coumaric acid belonged to this group. Thus, these components are not only free of acute and chronic toxicity to *C. elegans*, but they also increase the growth, motility, regeneration, and numbers. Additional benefits of these components have been reported in other studies as well [93,94,95,96,97]. A recent study by Land et al. showed increased motility and an increase in late-stage life activity by up to 206% when *C. elegans* were treated with CBD compared with the control group, and no animal died when exposed to 0.4–4000 μM CBD [93]. Tea tree (*Camellia tenuifolia*) oil has been reported as an antioxidant agent [94]. Coumaric acid is reported as an antioxidant agent, ameliorates oxidative and osmotic stress, and provides lifespan extension, and anti-ageing in the *C. elegans* model [95,96,97].

## 4. Materials and Methods

### 4.1. Systematic Review for Selecting Highly Effective Plant-Based Natural Compounds (PBCs) as Antibacterial Agents

We conducted a systematic review of available publications to generate a list of potentially highly effective antibacterial plant-based natural compounds (PBCs). The detailed systematic review approach is provided in the Appendix A.

### 4.2. Bacterial Strains, Culture Media, Antibiotics, and Plant-Based Natural Compounds (PBCs)

The list of PBCs and antibiotics as well as the bacterial strains and culture media that were used in this study are shown in the Appendix A.

### 4.3. Antibacterial and Antibiofilm Assays

See the Appendix A for all antibacterial and antibiofilm assays.

### 4.4. Cannabis Sativa Oil Extraction

See the Appendix A for the cannabis sativa oil extraction method.

### 4.5. Hydrogen Peroxide Assay

The hydrogen peroxide concentration, after exposure with selected PBCs, ciprofloxacin, and gentamycin, was detected with the Pierce Quantitative Peroxide Assay Kit in the aqueous-compatible formulation according to the manufacturer’s instructions [98]. For preparing the standard, 1 mM solution of H_2_O_2_ was initially made by diluting a 30% H_2_O_2_ stock 1:9000 (11 μL of 30% H_2_O_2_ into 100 mL of double-distilled (DD) water). This sample was then serially diluted with DD water 1:2 (100 μL of DD water + 100 μL of the previous dilution) for a total of 11 samples as a standard. Then, 200 μL of the working reagent (WR) from the kit was added to 20 μL of the diluted H_2_O_2_ standards. Samples were mixed and incubated for 15 min at 21 °C in the dark. Absorbances were measured at 595 nm using a Thermomax microtiter plate reader with Softmax Pro data analysis software (Molecular Devices, Sunnyvale, CA).

For measuring the treated and untreated samples with PBCs, the bacteria were cultured in 3 mL of MHB and were incubated at 37 °C for ~3 h in a shaker incubator (150 rpm) to reach an OD600 of 0.08. Then, treated with MIC concentrations of agents and untreated groups with phosphate-buffered saline (PBS) (as a negative control), the positive control was treated with 250 μM H_2_O_2,_ and incubated at 37 °C for 1 h in a shaker incubator (150 rpm). Ciprofloxacin and gentamycin were used as the controls. The bacterial cells were washed with PBS by centrifuging (10,000 g for 5 min) and discarding the supernatant. Then, 3 mL PBS was added to each sample and vortexed, and 200 μL of the WR was added to 20 μL of each sample. Samples were mixed and incubated for 15 min at room temperature. Absorbances were measured at 595 nm using a Thermomax microtiter plate reader with Softmax Pro data analysis software (Molecular Devices, Sunnyvale, CA, USA). The blank value was subtracted from all sample measurements. The samples’ H_2_O_2_ concentrations were calculated based on the standard carve R^2^ = 0.93 value.

### 4.6. Iron Detection Ferene-S Assay

The release of Fe^2+^ from the iron–sulfur cluster in *P. aeruginosa* was detected using a Ferene-S assay with the probe, 3-(2pyridyl)-5,6-bis(2-(5-furylsulfonic acid))-1,2,4-triazin (Sigma-Aldrich, St Louis, MO, USA) [99]. Then, 10 mL of bacteria (OD600 of 0.08) was prepared in Tris-HCl buffer (20 mM, pH 7). The bacterial cells were washed with the same buffer by centrifuging (10,000× *g* for 5 min) and discarding the supernatant. The platelets (bacterial cells) were then lysed by sonication using a 250HT ultrasonic cleaner (VWR International) set at 60 Hz for 20 min in the same buffer. The samples were centrifuged (10,000× *g* for 5 min) and the supernatant was collected. The solution was treated with an MIC concentration of PBCs, the negative control (dd H_2_O), and the positive control (90 °C for 10 min), as well as ciprofloxacin and gentamycin as the internal control. Then, 10 mM Ferene-S probe was added to each sample in a 96-well plate, and samples were incubated at 21 °C in the dark for 1 h. Absorbance measured at 600 nm, using a Thermomax microtiter plate reader with Softmax Pro data analysis software (Molecular Devices, Sunnyvale, CA, USA) [100].

### 4.7. Membrane Disruption and Permeability Measurements

For the measurement of the membrane permeability, propidium iodide (PI) (Invitrogen, Eugene, Oregon, USA) was used as the fluorescent reporter dye. Increased PI fluorescence (read) is correlated with increased membrane disruption and permeability as it can enter the cells and bind to DNA [101]. The bacteria were cultured in 3 mL of MHB and were incubated at 37 °C for ~3 h in a shaker incubator (150 rpm) to reach the OD600 of 0.08. Each of the groups was treated with MIC concentrations of agents, PBS (as a negative control), and bacteria incubated at 90 °C for 10 min (as a positive control), and then incubated at 37 °C for 1 h in a shaker incubator (150 rpm). Ciprofloxacin and gentamycin were used for the internal control. The samples were centrifuged and washed with PBS (10,000× *g* for 2 min). The platelets were stained with 0.16 mM PI for 5 min at 21 °C in the dark, then 10 μL of the samples were transferred onto slides and examined on a fluorescence microscope (Zeiss axio imager Z1) at the same exposure time (640 ms red, 1 s green). Densitometry analysis was performed by Fiji software (ImageJ Version 1.51).

### 4.8. Toxicity Test by the Caenorhabditis elegans Model

*C. elegans* handling, growth and manipulation were performed using standard methods [102]. Initially, experiments were carried out using M9 liquid culture [103]. However, because the working solution of different components had different densities, it affected the worm’s motility in the liquid media. Therefore, for the prevention of bias, all experiments were transferred to NGM plates [103].

The *C. elegans* wild-type strain N2 (Bristol) and *E. coli* OP50 at 10^9^ CFU/mL as a food source were used in this study [102,103]. *E. coli* and 10 times the MIC of each antibacterial component were added to the agar plates. In total, 4–6 synchronized L2 larvae in 10 µL of phosphate-buffered saline (PBS) were transferred onto the plates, with three biological repeats for each condition (total n = 12–18), and the exact number of larvae for each plate were recorded. Plates without any PBCs were used as controls. Treated and untreated groups were incubated at 20 °C in the dark and results were obtained for five days. Four results were recorded each day: (1) the number of live worms; (2) motility (head swing and body-bending frequency per minute); (3) growth (body length and body width); (4) regeneration (generation of offspring, as well as size, number, and motility of them were considered) rates. The treated and untreated groups were compared for reporting the ratio of the variables and results.

### 4.9. Antioxidant Test by the Caenorhabditis elegans Model

ROS levels lead to oxidative stress, inflammation, and cellular disruption, while reduced glutathione (GSH) is a cellular antioxidant against oxidative stress, and it is essential to cellular protection [104]. Multiple fluorescent sensor probes were employed including dihydroethidium (DHE), 2′,7′-dichlorofluorescin diacetate (DCFH-DA), and naphthalene-2,3-dicarboxal-dehyde (NDA), all obtained from the Invitrogen (Carlsbad, CA, USA) to detect the O_2_^−^, ROS, and glutathione, respectively.

Different time frames and exposure times were examined to find the optimal conditions to see the differences between the test groups. After the treatment of L2 nematodes with 10× MIC concentration of each PBC and antibiotics in NGM plates at 20 °C, the O_2_^−^ and ROS levels were measured after 4 days, while glutathione levels were measured after 24 h. The plates were washed, and the liquid was centrifuged (6000× *g*) 2 times with PBS. The nematodes were exposed to DHE (20 μM), DCF (10 μM), and NDA (20 μM) probes to measure the O_2_^−^, ROS, and glutathione, respectively. The nematodes were incubated with probes at 20 °C for 4 h. The liquid was gently washed 2 times with PBS buffer and the animals were examined on a fluorescence microscope (Zeiss axio imager Z1) with identical exposure time (1.5 s). Densitometry and intensity analysis was performed by Fiji software (ImageJ).

### 4.10. Statistical Tests and Data Analysis

All data organization, mean, standard deviation, mode, analysis, and the three-dimensional graphical representations were performed using Microsoft Excel 365 (Microsoft Corporation, Redmond, WA, USA). The significance between the two groups was determined by a two-tailed Student’s *t*-test. All results are expressed as mean ± standard error. All experiments were repeated with at least three biological replicates and any experiments that had more variable results were repeated seven times.

## 5. Conclusions

Some of the PBCs tested, including THC, CBD, cinnamaldehyde, and carvacrol, showed quite promising antibacterial and antibiofilm potency in comparison with common antibiotics (ciprofloxacin and gentamicin). They are not only non-toxic but also have antioxidant properties as well. Our mechanism data suggest that they mostly target the bacterial cell membrane rather than using ROS or the oxidative stress pathway. These data showed that H_2_O_2_ and Fe^2+^ concentrations in our selected PBCs were almost the same or less than ciprofloxacin, gentamycin, and the untreated group. Therefore, such information will allow for knowledge-driven second-generation PBC antimicrobial development and logical partnering for synergistic efficacies. Animal model studies and the screening of synergism effects for selected PBCs are recommended for future studies.

## Figures and Tables

**Figure 1 antibiotics-11-01099-f001:**
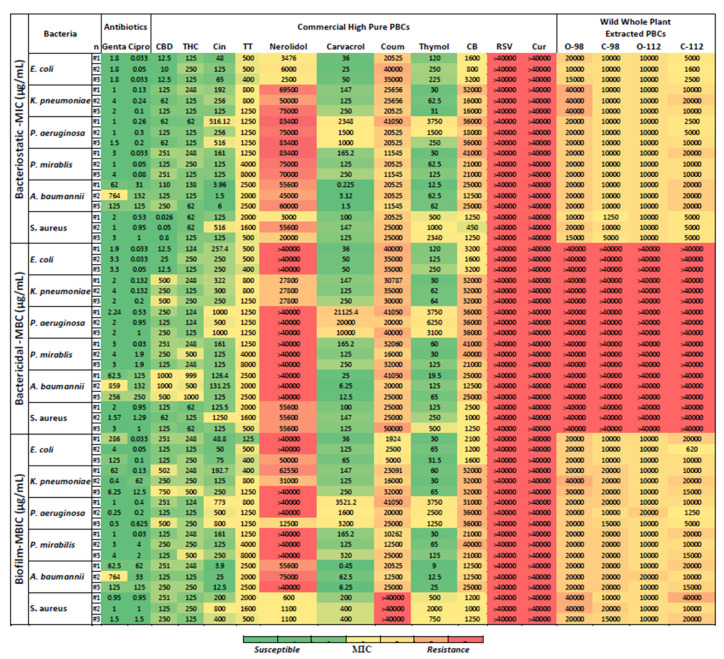
Minimum inhibitory concentration (MIC), minimum bactericidal concentration (MBC), and minimum biofilm inhibitory concentration (MBIC) of 15 different plant-based natural compounds (PBCs) against various pathogens of concern. Genta = gentamicin; Cipro = ciprofloxacin; Cin = cinnamaldehyde; TT = tea tree oil; Coum = coumaric acid; CB = Canada balsam; RSV = resveratrol; Cur = curcumin; n = number of trials.

**Figure 2 antibiotics-11-01099-f002:**
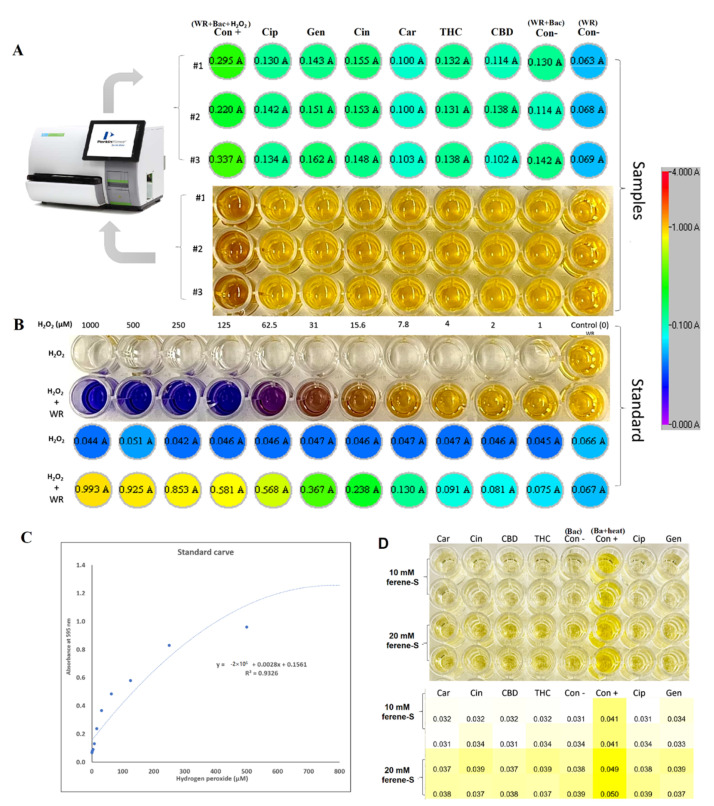
Hydrogen peroxide and free iron [(ferrous) Fe^2+^] concentration. (**A**) H_2_O_2_ concentrations in the samples with the naked eye (down) and plate reader (upper), with ciprofloxacin and gentamycin used as the internal controls. (**B**) Standard 1 mM solution of hydrogen peroxide serially diluted with double-distilled (DD) water 1:2 for a total of 11 samples. DD water was used as the blank working reagent (WR) as a negative control, and bacteria treated with 250 μM H_2_O_2_ as a positive control. (**C**) Standard curve after calculating standard curve based on standard concentrations and OD means; H_2_O_2_ concentrations of all samples were calculated based on R^2^ value. (**D**) Free iron [(ferrous) Fe^2+^] concentration in a *P. aeruginosa* lysate after 1 h treatment with selected PBCs. Bacterial incubated in 95 °C for 10 min to break the Fe-S complex as a positive control. The naked eye (upper) heatmap (down) is illustrated in the panels. Con+: control-positive (Bacteria + WR + H_2_O_2_); Con−: control-negative (Bacteria + WR); Cin: cinnamaldehyde; THC: tetrahydrocannabinol; CBD: cannabidiol; Cip: ciprofloxacin; Gen: gentamycin.

**Figure 3 antibiotics-11-01099-f003:**
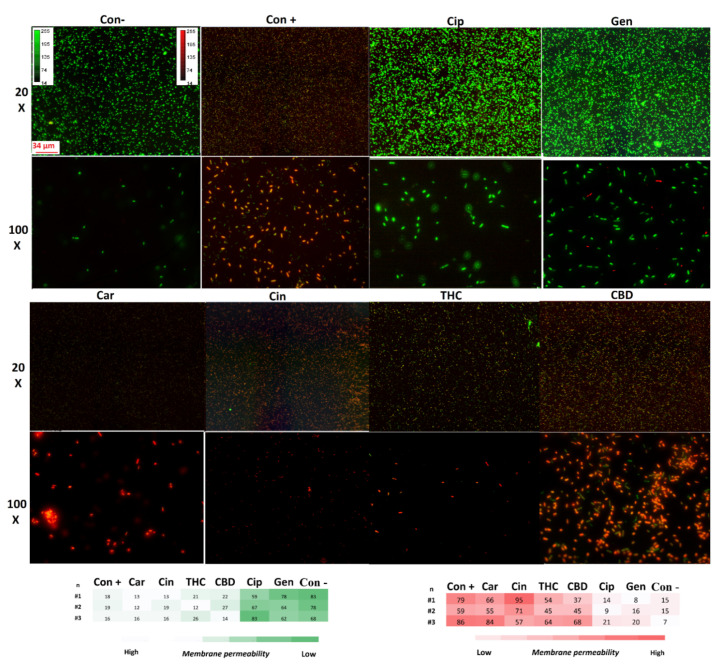
The fluorescence microscopy of PI-staining of *P. aeruginosa* exposed with selected PBCs, antibiotics, and controls for 1 h. Higher red fluorescence has higher membrane disruption and permeability. The densitometry and intensity measurement heat map is shown in the bottom panel (*n* = 3). Con+: control-positive (bacteria boiled for 10 min to disrupt the membrane); Con−: control-negative (untreated); Cin: cinnamaldehyde; THC: tetrahydrocannabinol; CBD: cannabidiol; Cip: ciprofloxacin; Gen: gentamycin.

**Figure 4 antibiotics-11-01099-f004:**
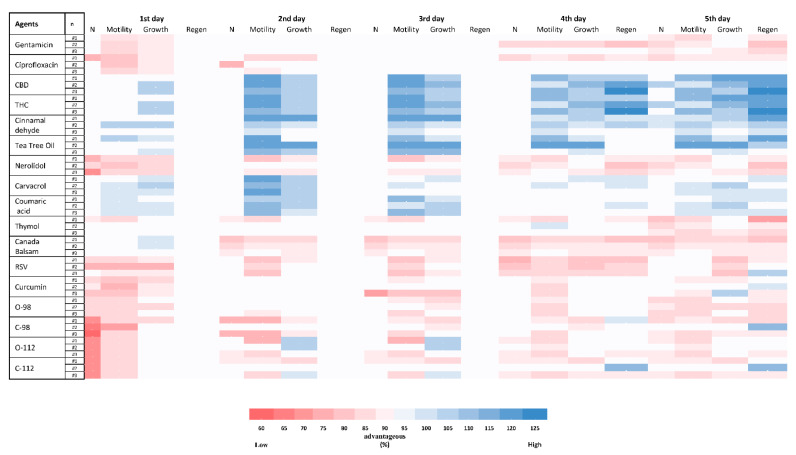
Heatmap on survival ratio of *C. elegans* and toxicity of the 15 different plant-based natural compounds (PBCs) as an antibacterial agent in the *C. elegans* model. All treated groups with PBCs were compared with the untreated group and the results are reported in the ratio (%). The red colour shows the lower advantage and the groups with the blue colour had the higher advantage in comparison with the control group. n = number of trials; N = number of animals; Regen = regeneration.

**Figure 5 antibiotics-11-01099-f005:**
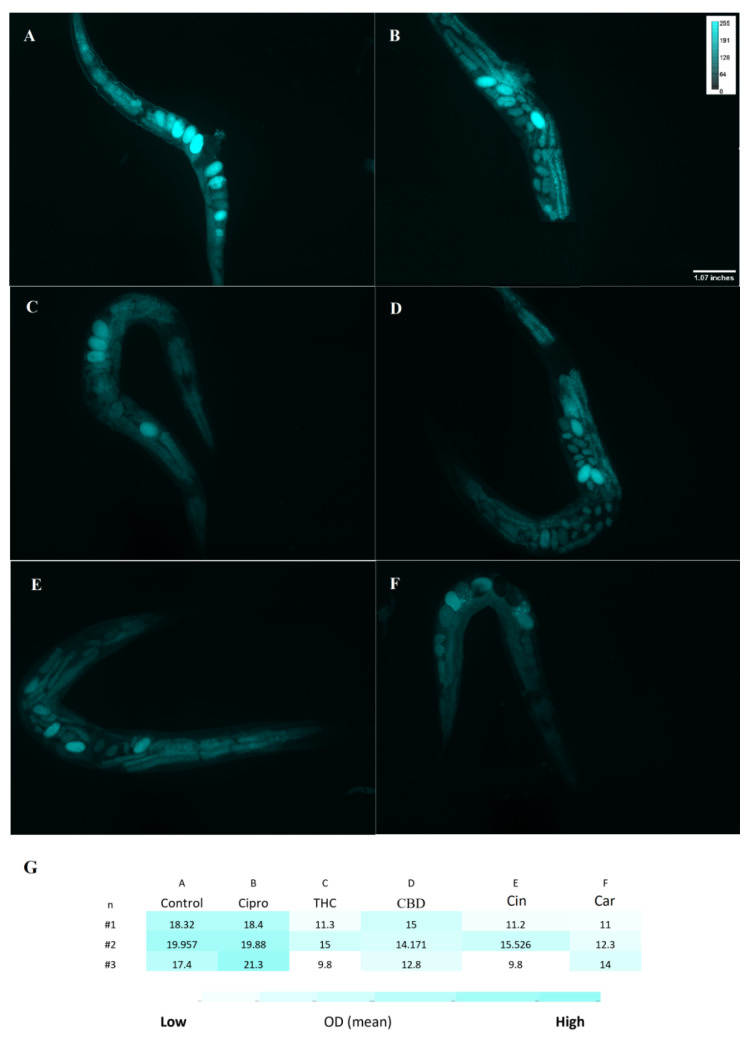
DCFH-DA staining for reactive oxygen species (ROS) level. Fluorescence intensity of all *C. elegans* ROS levels was measured after exposure to control (PBS) (**A**), ciprofloxacin (**B**), THC (**C**), CBD (**D**), cinnamaldehyde (**E**), and carvacrol (**F**) by DCFH-DA probe. Heatmap on ROS level of *C. elegans* in the treated and untreated groups showed in panel (**G**). n = trial number.

**Figure 6 antibiotics-11-01099-f006:**
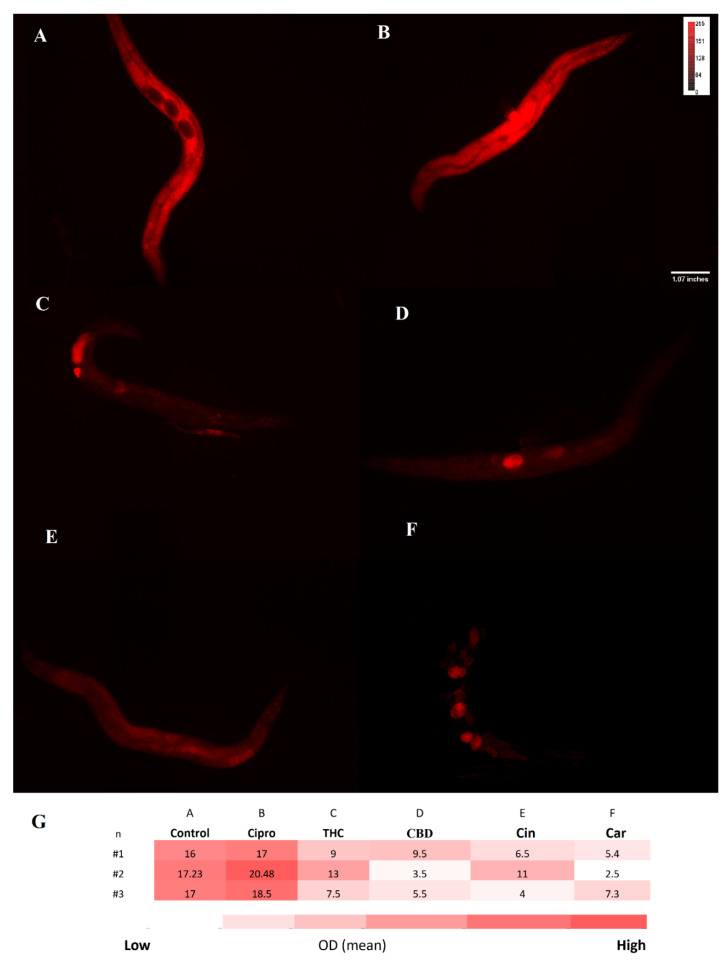
DHE staining for O_2_^−^ level. Fluorescence intensity of all *C. elegans* O_2_^−^ levels was measured after exposure to control (PBS) (**A**), ciprofloxacin (**B**), THC (**C**), CBD (**D**), cinnamaldehyde (**E**), and carvacrol (**F**) by DHE probe. Heatmap on O_2_^−^ level of *C. elegans* in the treated and untreated groups shown in panel (**G**). n = trial number.

**Figure 7 antibiotics-11-01099-f007:**
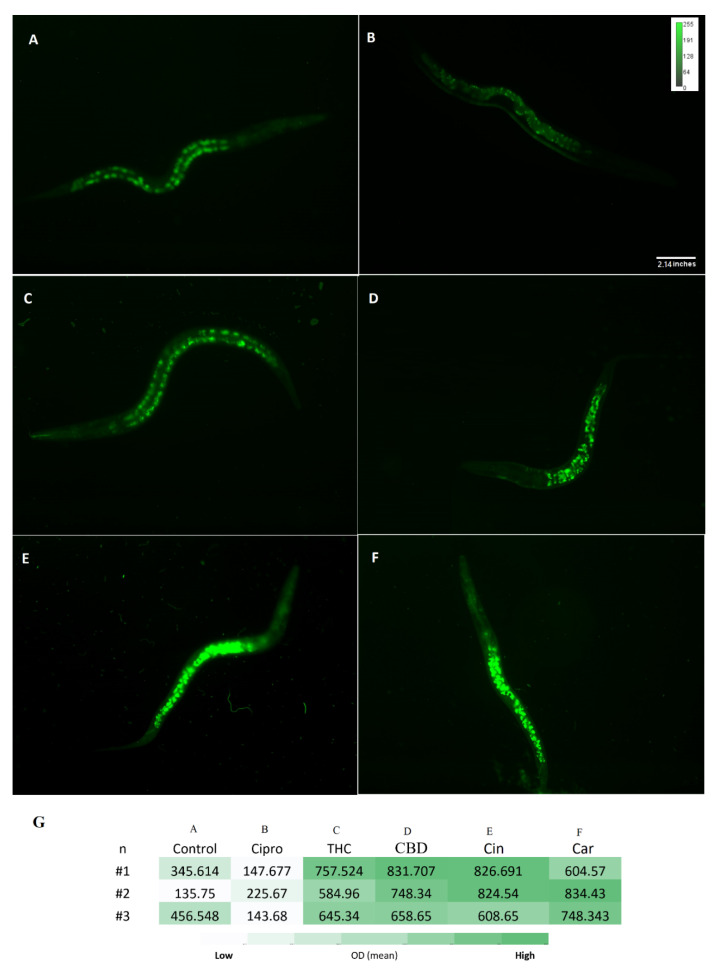
NDA staining for reduced glutathione level. Fluorescence intensity of all *C. elegans* GSH levels was measured after exposure to control (PBS) (**A**), ciprofloxacin (**B**), THC (**C**), CBD (**D**), cinnamaldehyde (**E**), and carvacrol (**F**) by NDA probe. Heatmap on glutathione level of *C. elegans* in the treated and untreated groups shown in panel (**G**). n = trial number.

**Figure 8 antibiotics-11-01099-f008:**
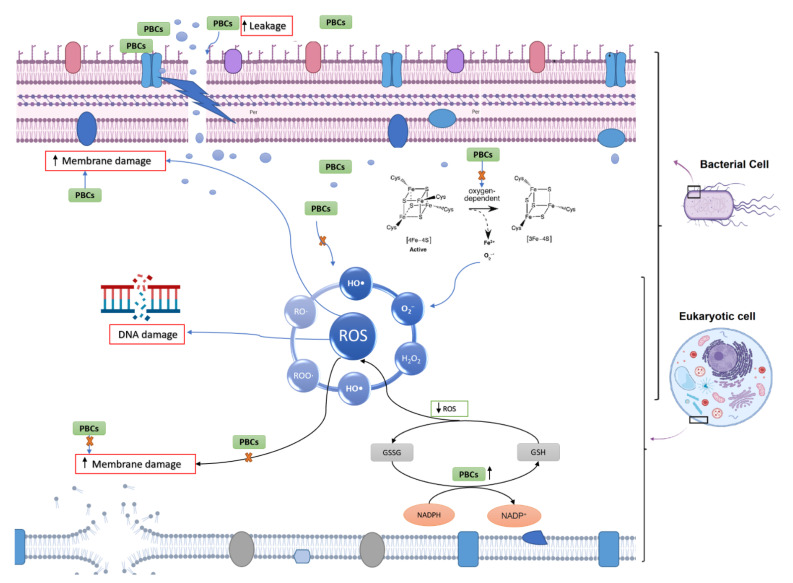
Mechanism illustration of plant-based components (PBCs) in bacterial and eukaryotic cells. Overall, PBCs target the bacterial cell membrane (but not eukaryotic cells) because of their different membrane structure. GSH = glutathione; GSSG = oxidized glutathione.

**Table 1 antibiotics-11-01099-t001:** The most effective antibiofilm and antibacterial PBCs for targeted treatment towards defined pathogen species.

Bacteria	The Most Effective Antibiofilm and Antibacterial PBCs			
	PBCs	MIC (μg/mL)	MBC (μg/mL)	MBIC (μg/mL)
*S. aureus*	CBD	0.026–0.8	62–125	125–150
	THC	62–125	62–125	125–150
*A. baumannii*	Cinnamaldehyde	1.5–3.96	126.5–131.25	4–12.5
	Carvacrol	0.225–3.12	6.25–25	1–62.5
	Thymol	12.5–62.5	19.5–125	9–25
*K. pneumoniae*	Thymol	30–62.5	30–64	30–65
*E. coli*	CBD	10–12.5	12.5–25	125–250
	Carvacrol	25–50	36–50	36–125
*P. aeruginosa*	CBD	62–125	125–250	125–500
	THC	62–125	125	125–250
*P. mirabilis*	Thymol	30–125	60–125	30–125

Minimum inhibitory concentration (MIC), minimum bactericidal concentration (MBC), and minimum biofilm inhibitory concentration (MBIC) of cannabidiol.

## Data Availability

The data presented in this study are available on request from the corresponding author.

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
