# Peer review of "Antibacterial, Antibiofilm, and Antioxidant Activity of 15 Different Plant-Based Natural Compounds in Comparison with Ciprofloxacin and Gentamicin"

_antibiotics, 2022, doi:10.3390/antibiotics11081099_

Round 1
Reviewer 1 Report
This review is well written and structured, only few more details:
In Introduction (row 58), please specify WHO priority listed bacteria, regarding their antimicrobial resistance.
For the Conclusions section, only few more words on the prospects for future research in this topic.
Author Response
Reviewer1
Point 1: In Introduction (row 58), please specify WHO priority listed bacteria, regarding their antimicrobial resistance.
Response: We have added a few brief sentences to add clarification.
“for the Gram-negative bacteria Pseudomonas. aeruginosa, Escherichia coli, Klebsiella pneumoniae, Proteus mirabilis, Acinetobacter baumannii, and the Gram-positive, Staphylococcus aureus”
Point 2: For the Conclusions section, only few more words on the prospects for future research in this topic.
Response: Thanks for pointing this out, we are agreement with your comment and we have added a brief explanation.
“Animal model studies and screening synergism effects of selected PBCs is recommended for future direction”
Reviewer 2 Report
1. Out of 15 PBC? or 11 PBCs.
2. Over the past 10 years, researchers have explored different strategies and advanced 37 technologies to tackle the problem. - this sentence seems to be standing alone' not connecting with the earlier sentence.
3. approaches to formulate novel antibiotics? can be antimicrobials rater antibiotics?
4. as well as to two common antibiotics?
5. The antimicrobial challenge can lead to membrane damage and decomposition of respiratory enzymes leading to breaking down [Fe-S] clusters which release Fe+2 which can catalyze Fenton reactions towards ROS productions, is too hard to understand?
6. The authors should avoid repeating sentences.
7. Typo errors and English grammar should be corrected.
Author Response
Reviewer 2
Point 1: Out of 15 PBC? or 11 PBCs.
Response: Thanks for pointing this out. Out of 15 is correct, because we had 4 of our own extracts in addition to the specific plant-based components. So in total 15 PBCs. This has been clarified in the introduction.
Point 2: Over the past 10 years, researchers have explored different strategies and advanced 37 technologies to tackle the problem. - this sentence seems to be standing alone' not connecting with the earlier sentence.
Response: Thanks for pointing this out, we are agreement with your comment and we deleted the sentence.
Point 3: approaches to formulate novel antibiotics? can be antimicrobials rater antibiotics?
Response: We are agreement with your comment and we revised the sentence.
Point 4: as well as to two common antibiotics?
Response: We have added a brief explanation to clarify the sentence.
“we compared their antimicrobial and antibiofilm potency with two common antibiotics (ciprofloxacin and gentamicin)”
Point 5: The antimicrobial challenge can lead to membrane damage and decomposition of respiratory enzymes leading to breaking down [Fe-S] clusters which release Fe+2 which can catalyze Fenton reactions towards ROS productions, is too hard to understand?
Response: We revised to clarify the sentence.
“The antimicrobial challenge can lead to ROS productions31-33. This can occur from: 1. membrane damage, that results in uncoupling of the electron transport chain and thus increases electron flow to oxygen; 2. decomposition of respiratory enzymes leading to breaking down [Fe-S] clusters and subsequent release of Fe+2 which can catalyze Fenton reactions.”
Point 6: The authors should avoid repeating sentences.
Response: We carefully read the text and revised the repeated sentences to avoid repetition.
Point 7: Typo errors and English grammar should be corrected.
Response: The minor mistakes have been checked and corrected.
Reviewer 3 Report
lines 16 and 61 (and paragraph 1.7 of SI): "strains" of Cannabis? maybe "cultivar", "ecotype"...
line 24: remove "the"
lines 152-156: in my opinion if a compound has a percentage higher than the untreated control (produced), but if the percentage is lower (reduced)
lines 166 and 386: correct "concertation"
lines 254-255: Reported....reported
line 274: planet?
text and SI (including figures and tables): K. pneumoniae not K. pneumonia and P. mirabilis not P. miriablis
line 290: advantage of them?
lines 292 and 302: the mechanism of carvacrol and thymol has been recently proposed as related to carbonic anhydrase inhibition in different bacteria (doi: 10.3390/ijms222111583.)
lines 360 and 371: change to "were added"
line 369: comparitor?
lines 370, 382, 385, 400, 434: convert rpm in g
line 379: correct the chemical name of the compound
line 382: change to "The platelets were"
Figure 2 caption, line 474: correct "gentamycine"
line 480: correct "Bacterial boiled"
lines 294-295: Different...different
Figure 8 caption line 514: correct "PBCs is"
some refs are not correctly written (refs 103 and 104)
Supporting information paragraph 1.3: the product code is useful but the purity must be expressed
correct in Supporting information and tables/Figures: O-coumaric. It should be o-coumaric
paragraph 1.4: change to "were than added"
paragraph 1.5: change to "were transferred"
Supporting information: Cannabis sativa must be italicized
paragraph 1.7: the CBD and THC concentrations must be provided
paragraph 1.6 and related results: how is it possible that the authors consider values higher than MBC as MBIC and not MBEC values? If at lower concentrations (MBC) bacteria are killed how could they produce biofilm? This is not biofilm prevention. Usually MBIC values are lower than MICs. Values higher than MICs can be used to calculate mature biofilm eradication (MBEC). This point is the mjor concern of this article.
Author Response
Reviewer 3:
Point 1: lines 16 and 61 (and paragraph 1.7 of SI): "strains" of Cannabis? maybe "cultivar", "ecotype"...
Response: yes should be ‘cultivar’ and corrected.
Point 2: line 24: remove "the"
Response: Revised and thank you.
Point 3: lines 152-156: in my opinion if a compound has a percentage higher than the untreated control (produced), but if the percentage is lower (reduced)
Response: Thanks for pointing this out, we are agreement with your comment and we revised the sentence.
Point 4: lines 166 and 386: correct "concertation"
Response: Thanks for pointing this out, we revised them.
Point 5: lines 254-255: Reported....reported
Response: We deleted the extra “reported”.
Point 6: line 274: planet?
Response: Thanks for pointing this out, revised.
Point 7: text and SI (including figures and tables): K. pneumoniae not K. pneumonia and P. mirabilis not P. miriablis
Response: Thanks for pointing this out, revised.
Point 8: line 290: advantage of them?
Response: We deleted the extra “advantage of them”.
Point 9: lines 292 and 302: the mechanism of carvacrol and thymol has been recently proposed as related to carbonic anhydrase inhibition in different bacteria (doi: 10.3390/ijms222111583.)
Response: We are aware of this study. It is curious to consider that since CA is thought to be involved in decreasing H2O2 levels, if inhibited by carvacrol, why do we see reduced H2O2 carvacrol addition? There are mechanisms proposed for many of the compounds we studied, with unfortunately, contradictory conclusions from different groups. Thus in part our reasoning for this review was a direct comparison between the ‘essential oils’ beside each other in the same conditions. We chose purposely not to dwell on proposed mechanisms. Thus, in our meta- analysis we only chose to address the antimicrobial activity reported and not the biochemical mechanisms suggested in the studies. To drill down on mechanisms a full systematic study should be performed with Biochemical assays under identical conditions to defined strains. No change in the text is provided.
Point 10: lines 360 and 371: change to "were added"
Response: Thank you and revised.
Point 11: line 369: comparitor?
Response: Thank you, revised.
Point 12: lines 370, 382, 385, 400, 434: convert rpm in g
Response: Thank you, revised.
Point 13: line 379: correct the chemical name of the compound
Response: Thank you, the name is correct.
Point 14: line 382: change to "The platelets were"
Response: Thank you, revised.
Point 15: Figure 2 caption, line 474: correct "gentamycine"
Response: Thank you, revised.
Point 17: line 480: correct "Bacterial boiled"
Response: Thank you, revised.
Point 18: lines 294-295: Different...different
Response: Thank you, revised.
Point 19: Figure 8 caption line 514: correct "PBCs is"
Response: Thank you, revised.
Point 20: some refs are not correctly written (refs 103 and 104)
Response: Thank you, revised.
Point 21: Supporting information paragraph 1.3: the product code is useful but the purity must be expressed
Response: the purity of each components added.
Point 22: correct in Supporting information and tables/Figures: O-coumaric. It should be o-coumaric
Response: Thank you, revised.
Point 23: paragraph 1.4: change to "were than added"
Response: Thank you, revised.
Point 24: paragraph 1.5: change to "were transferred"
Response: Thank you, revised.
Point 25: Supporting information: Cannabis sativa must be italicized
Response: Thank you, revised.
Point 26: paragraph 1.7: the CBD and THC concentrations must be provided
Response: Herer, in this study the crude exacted concentration of CBD and THC was not determined and the company providing would not provide the relative amounts or %. It would be good in the future to compare various extraction methods from various plant materials, but our legal use license of this controlled substance did not allow such detailed studies. We have added a clear sentence.
The relative concentrations of CBD and THC in the resulting oils were not known or determined and information not provided by the supplier.
Point 27: paragraph 1.6 and related results: how is it possible that the authors consider values higher than MBC as MBIC and not MBEC values? If at lower concentrations (MBC) bacteria are killed how could they produce biofilm? This is not biofilm prevention. Usually MBIC values are lower than MICs. Values higher than MICs can be used to calculate mature biofilm eradication (MBEC). This point is the mjor concern of this article.
Response: Thanks for pointing this out. MBIC depends on incubation time, if we incubated in 24 h, one may see MBIC values equal or less than MIC (but for some species this is not observed). For most strains studied here 24 h is not long enough to see significant amount of biofilm, so we incubate them for 48 h. In the longer incubation, bacteria keep growing (albite slow if only bacteriostatic, even in the higher concentration of MIC), so MBIC values can be greater than MIC, which we find to be typical for many non-traditional antibiotics. Sentence added in this regard to results section on MBIC.
Although MBIC is expected to be similar to the MIC as seen with most antibiotics, for many non-antibiotic antimicrobials and particularly those that are bacterial static versus bactericidal, the MBIC can be quite different from the MIC.
Round 2
Reviewer 2 Report
line no: 59: Pseudomonas. aeruginosa? a full stop should be removed
Reviewer 3 Report
the author have considered my issues correctly